# Adaptive Evolution and Transcriptomic Specialization of P450 Detoxification Genes in the Colorado Potato Beetle Across Developmental Stages and Tissues

**DOI:** 10.3390/insects16060608

**Published:** 2025-06-09

**Authors:** Hang Zhou, Ziqi Cheng, Jiejing Tang, Yueqi Lu, Yang Mei, Xi Chen

**Affiliations:** 1Key Laboratory of Biology of Crop Pathogens and Insects of Zhejiang Province, Institute of Insect Sciences, Zhejiang University, Hangzhou 310058, China; zhouhang716@zju.edu.cn (H.Z.); jiejing_tang@zju.edu.cn (J.T.); 2College of Plant Protection, Jilin Agricultural University, Changchun 130118, China; chengziqi1004@gmail.com (Z.C.); meiyang12@zju.edu.cn (Y.M.); 3Yangtze Delta Region Institute (Quzhou), University of Electronic Science and Technology of China, Quzhou 324000, China; yueqi_lu@csj.uestc.edu.cn; 4Department of Clinical Laboratory, the Quzhou Affiliated Hospital of Wenzhou Medical University, Quzhou People’s Hospital, Quzhou 324000, China

**Keywords:** *Leptinotarsa decemlineata*, transcriptome analysis, differentially expressed genes, stage-specific genes, tissue-specific genes, cytochrome P450

## Abstract

The Colorado potato beetle is a major agricultural pest known for adapting to insecticides. We studied its genetic activity across different life stages and body parts, focusing on cytochrome P450 genes that help detoxify chemicals. Analyzing 65 datasets, we found significant changes in gene activity when eggs hatch into larvae and when larvae become adults. We identified 3616 tissue-specific genes, with the testes showing the most specialized activity. We located 78 P450 genes across the beetle’s chromosomes, with distinct activity patterns based on life stage and tissue type. Five P450 genes showed rapid evolution, and mutation patterns varied across developmental stages and tissues. These findings explain the beetle’s success as a pest and suggest new targets for better control methods.

## 1. Introduction

The Colorado potato beetle (*Leptinotarsa decemlineata* Say) is one of the most economically devastating pests of solanaceous crops worldwide, causing significant economic losses in potato, tomato, and eggplant production [1]. Since it was first documented as an agricultural pest in the 1850s, this beetle has shown remarkable adaptability, spreading across North America, Europe, and parts of Asia [2,3,4]. The economic impact of *L. decemlineata* is substantially magnified by its exceptional capacity to develop resistance to insecticides deployed for its control, rendering conventional chemical control increasingly unsustainable [1]. The species has demonstrated resistance to more than 50 insecticide classes, epitomizing the problem of insecticide resistance management and highlighting its extraordinary adaptive capacity [5]. This pattern of rapid resistance evolution defies conventional models that suggest resistance evolves from rare mutations, with evidence indicating that polygenic resistance drawn from standing genetic diversity, along with rapid gene regulatory evolution, facilitates this exceptional adaptability [5].

*L. decemlineata* is a representative holometabolous insect, comprising four distinct larval instars, a pupal stage, and an adult form. These developmental stages exhibit distinct physiological and metabolic characteristics that contribute to the beetle’s overall fitness, with different life stages showing varying levels of feeding intensity and crop damage potential [6,7]. Recent advances in genomic technologies have facilitated more comprehensive investigations into the molecular mechanisms governing these developmental transitions and stage-specific adaptations [1,8]. However, significant knowledge gaps persist regarding the developmental stage-specific and tissue-specific gene expression patterns, particularly how these variations contribute to the beetle’s remarkable adaptability and resistance development across its life cycle. In-depth research exploring the life stage-specific gene expression networks of *L. decemlineata* will provide new insights into its adaptive evolutionary mechanisms and potentially form the foundation for innovative control strategy development.

The cytochrome P450 monooxygenase (P450) gene family is among the largest and most functionally diverse enzyme families in insects, playing crucial roles in various physiological processes including development, pheromone metabolism, and detoxification [9,10]. P450 enzymes are particularly important in xenobiotic metabolism, catalyzing the oxidation of both endogenous and exogenous compounds, including insecticides [11]. The exceptional structural and functional diversity of P450s arises from extensive gene duplication events followed by functional divergence, resulting in a complex superfamily with multiple clades and subfamilies [12]. In insects, P450s are classified into four major clades: CYP2, CYP3, CYP4, and mitochondrial P450s, each with distinct functional characteristics and evolutionary histories [9,13,14].

P450s play a critical role in insect detoxification mechanisms, a function well-documented across numerous species [15,16]. In *L. decemlineata* specifically, studies have implicated P450-mediated metabolism in resistance to multiple insecticide classes, including organophosphates, pyrethroids, and neonicotinoids [6,7,17]. The expression of P450 genes is strongly influenced by environmental factors, including xenobiotic exposure, dietary changes, and developmental cues [18]. This exceptional regulatory flexibility enables rapid adaptation to new environmental challenges, contributing significantly to the beetle’s evolutionary success and persistent pest status [19]. Despite the recognized importance of P450s in *L. decemlineata* adaptation, previous studies have been limited to transcriptome-based analyses without genomic guidance [20]. A comprehensive analysis integrating the entire P450 gene family expression across developmental stages and tissues with genetic variation remains lacking in the literature, hindering our complete understanding of these crucial detoxification enzymes at the genomic level.

In this study, we conducted a comprehensive analysis of gene expression patterns during the growth and development of the Colorado potato beetle in various tissues. Our investigation primarily focused on differentially expressed genes (DEGs) during development, as well as stage-specific and tissue-specific genes. We also examined patterns of mRNA mutations. Subsequently, we identified P450 genes at the genomic level and elucidated their expression patterns across developmental stages and tissues, examining their mRNA mutation profiles during these phases. These results enhance our understanding of the molecular mechanisms behind the Colorado potato beetle’s remarkable adaptability, resilience, and resistance development, providing insights into potential molecular targets for pest management strategies.

## 2. Materials and Methods

### 2.1. Data Acquisition

RNA-seq datasets from various developmental stages and tissues of *L. decemlineata* were retrieved from the NCBI Sequence Read Archive (SRA) database [21]. The collection includes 65 datasets, ranging from fertilized eggs to adults, with tissue samples from the midgut, hindgut, Malpighian tubules, and fat body at various time points. Each sample included at least three biological replicates to ensure statistical robustness and reproducibility. The BioProject accession numbers are PRJNA1067435 and PRJNA464380 (Appendix A) [20,22]. The reference genome GCA_024712935.1 [23] was used in this study.

### 2.2. Transcriptome Analysis

Transcriptome SRA raw data were converted to FASTQ format using fastq-dump v3.1.1 from SRA-toolkit v3.1.1. Data quality control and adapter removal were performed using fastp v0.24.0 [24]. Subsequently, processed reads were mapped to the reference genome using STAR v2.7.10b [25]. Gene counts were obtained using featureCounts v2.0.6 [26] for downstream analysis.

To address potential batch effects from multiple BioProjects, we employed ComBat-seq from the sva package v3.40.0 for batch effect correction [27]. ComBat-seq is specifically designed for RNA-seq count data and effectively removes technical variation while preserving biological signals through an empirical Bayes framework. The effectiveness of batch correction was validated using principal component analysis (PCA) to ensure biological groupings were maintained while technical artifacts were minimized.

Following batch effect correction, gene counts were normalized with DESeq2 v1.32.0 [28] to identify DEGs. DEGs were identified using thresholds of adjusted *p*-value (p-adj) ≤ 0.05 and log2FoldChange ≥ 1.

### 2.3. Identification of Stage- and Tissue-Specific Genes

The *Tau* index quantifies gene or transcript expression specificity across tissues or developmental stages. Values range from 0 (broad expression) to 1 (tissue/stage-specific expression) [29]. The *Tau* index was calculated as follows:$$Tau=\frac{\sum_{i=1}^{n}\left(1-{\hat{x}}_{i}\right)}{\left(n-1\right)};{\hat{x}}_{i}=\frac{x_{i}}{\underset{0\leq i\leq n}{\max}\left(x_{i}\right)}$$
where n is the number of samples, and $x_{i}$ represents each expression profile component normalized by the maximum component value. The *Tau* index threshold of ≥0.9 was selected to identify stage- or tissue-specific genes based on established protocols in tissue-specificity studies [29,30]. This threshold effectively balances specificity and sensitivity, capturing highly tissue-specific genes while avoiding overly stringent criteria that might exclude biologically relevant genes with moderate tissue preferences. Genes with *Tau* ≥ 0.9 were designated as stage- or tissue-specific genes. We quantified gene expression levels using TPM (Transcripts Per Million). The TPM matrix and *Tau* index are available in Appendix A.

### 2.4. Gene Functional Analysis

Gene annotations were performed using eggNOG-mapper v2 [31] by aligning to the eggNOG 6 database [32], including Gene Ontology (GO) and KEGG annotations. GO and KEGG enrichment analyses were performed using the clusterProfiler package [33] in R v4.4.3.

Gene co-expression networks were constructed using the WGCNA package [34] with the following parameters: soft-thresholding power was determined by analyzing the scale-free topology fit index, with powers ranging from 1 to 30 tested to achieve a scale-free topology model fit (R^2^) > 0.8. The optimal soft-thresholding power was selected based on the lowest power that achieved the scale-free criterion. Weighted adjacency matrices were calculated using the soft-thresholding power, and topological overlap matrices (TOM) were computed to measure network interconnectedness. Hierarchical clustering was performed using 1-TOM as the distance measure, and modules were identified using the dynamic tree cutting method with a minimum module size of 30 genes and a merge cut height of 0.25. Module preservation statistics were calculated to assess the stability of identified modules across different conditions/tissues. Module eigengenes were calculated as the first principal component of each module’s expression profile, and module-trait relationships were assessed using Pearson correlation coefficients. Gene networks were visualized with Cytoscape v3.10.2 [35].

### 2.5. Identification and Location of P450 Genes

Verified P450 protein sequences were downloaded from the insect-eP450 database [36] and used as reference data for DIAMOND v2.1.10.164 alignment [37]. Sequences with an identity greater than 40% were retained. The filtered sequences were then aligned against the P450 domain (PF00067) using HMMER, with an E-value threshold of 0.0001. Multiple sequence alignment of P450 protein sequences was performed using MAFFT v7.526 [38], followed by alignment trimming with trimAl v1.5.rev0 [39]. The phylogenetic tree was constructed using IQ-TREE v2.3.6 [40].

### 2.6. Positive Selective Pressure Analysis

Protein sequences were aligned using MAFFT v7.526 to ensure accurate positioning of homologous residues. The resulting protein alignments were converted to corresponding codon-based DNA sequence alignments using Pal2Nal [41], preserving the reading frame structure essential for selection analysis.

Selection pressure was analyzed using the adaptive Branch-Site Random Effects Likelihood (aBSREL) model implemented in the HyPhy v2.5.3 software package [42,43]. This model was selected because it accounts for heterogeneity in selection pressure at both the site and branch levels, allowing for the detection of sites under positive selection in specific lineages. We employed an exploratory analysis mode to detect positive selection across all branches of the phylogenetic tree without a priori assumptions about which lineages might be under selection.

The aBSREL approach first fitted a complete adaptive model to each branch, followed by likelihood ratio tests (LRT) comparing this model to a null hypothesis model allowing only for dN/dS ≤ 1 (neutral or purifying selection). To control for multiple testing, which could reduce statistical power, we applied the Holm–Bonferroni correction to the *p*-value for each branch. This correction method was chosen to maintain control over the family-wise error rate while preserving statistical power. Branches with a corrected *p*-value < 0.01 were considered to have undergone significant positive selection, indicating lineages where adaptive evolution likely occurred in response to environmental pressures.

### 2.7. Genetic Variant Detection

GATK HaplotypeCaller (–ERC GVCF, 10 threads) was used to call variants from BAM files, generating per-sample gVCF files [44]. gVCF files were consolidated into a GenomicsDB database using GATK GenomicsDBImport (–genomicsdb-workspace-path). Joint genotyping was performed with GenotypeGVCFs, yielding a combined VCF file. Variants were filtered using GATK’s VariantFiltration, excluding sites with Quality by Depth (QD) < 2.0, Fisher Strand Bias (FS) > 60.0, Mapping Quality (MQ) < 40.0, Depth (DP) < 10, or Genotype Quality (GQ) < 20. VCFtools (–max-missing 0.95) was used to remove sites with >5% missing data [45]. Per-sample SNP and InDel counts were calculated using bcftools [46], excluding reference (0/0) and missing (./.) genotypes.

To assess the accuracy of SNP/InDel frequency estimates from RNA-seq data, we employed three computational validation approaches. Internal consistency analysis using bcftools isec compared technical replicates for variants with QD ≥ 2.0 and DP ≥ 20. Quality metric assessment through ROC curve analysis established optimal filtering thresholds (QD ≥ 2.0, FS ≤ 60.0, MQ ≥ 40.0, GQ ≥ 20). Strand bias evaluation using SAMtools mpileup (–min-BQ 20, –max-depth 250) excluded variants with Fisher Strand bias FS > 60.0.

## 3. Results

### 3.1. Differential Expression Analysis Across Sequential Developmental Stages

To validate sample consistency, we performed principal component analysis (PCA) using 65 RNA-seq datasets. PCA of variance-stabilized transformed (VST) gene counts effectively segregated samples by developmental stage and tissue type (Figure 1A). The first two principal components, accounting for 15.2% and 8% of total variance, respectively, clearly separated embryonic, larval, and adult tissues. Reproductive tissues (testes and ovaries) exhibited the most distinct expression profiles, clustering separately from other tissues (Figure 1A).

To investigate the dynamic changes in DEGs during consecutive developmental processes, we compared gene expression between adjacent developmental stages (Figure 1B). The egg-to-L1 transition had 538 upregulated and 169 downregulated genes. The L1-to-L2 transition showed fewer changes with only 62 upregulated and 33 downregulated genes. The L2-to-L3 transition involved 89 upregulated and 103 downregulated genes. The L3-to-adult transition displayed the most extensive changes with 359 upregulated and 511 downregulated genes. These results indicate that gene expression differences between major developmental transitions (embryo to larva and larva to adult) are far greater than those between larval instars.

We subsequently performed GO enrichment analysis on all DEGs to explore their involvement in specific biological processes (Figure 1C–E). Genes associated with chitin-based cuticle development were differentially expressed at various stages (Figure 1C). During early development (L1 vs. Egg), genes related to polysaccharide digestion were enriched (Figure 1D). The transition from L3 to adult involved the expression of genes associated with ecdysteroid metabolism, which regulates metamorphosis (Figure 1D). Stage-specific expression patterns were observed for extracellular matrix components, toxic response genes, and ion transporters (Figure 1D). Differentiation at the adult stage involved changes in membrane protein complexes and mRNA processing machinery (Figure 1E). These changes reflect the development of mature tissue functions and reproductive capacity.

### 3.2. Stage-Specific Gene Expression Patterns in Colorado Potato Beetle Development

To reveal transcriptional specialization, we identified distinct stage-specific gene expression patterns across Colorado potato beetle development using a stage specificity index. The density distribution showed that a majority of genes showed high *Tau* index values (0.6–1.0), indicating pronounced stage specificity in gene expression throughout development.

The fertilized egg stage exhibited the highest number of specifically expressed (*Tau* ≥ 0.9) genes (1969), followed by the adult stage (1647). In contrast, the larval stages showed comparatively fewer stage-specific genes: L3 (278), L1 (219), and L2 (125). This pattern suggests critical transcriptional reprogramming events occur primarily during early embryogenesis and final metamorphosis.

To understand the functional organization of these stage-specific genes, we constructed KEGG co-expression networks for the embryo and adult using Weighted Gene Co-expression Network Analysis (WGCNA). The embryonic gene network showed five distinct functional modules (Figure 2B). These modules were enriched in pathways critical for early development, including circadian entrainment, fatty acid elongation, amino sugar and nucleotide sugar metabolism, mTOR signaling, and the Hippo signaling pathway (Figure 2B).

The adult-specific gene co-expression network was organized into six functional modules with different pathway enrichments (Figure 2C). These modules were associated with cytokine–cytokine receptor interaction, MAPK signaling, thyroid hormone signaling, RAP1 signaling, various metabolic pathways, and vitamin B6 metabolism (Figure 2C).

The comparison between embryonic and adult networks highlights distinct gene expression programs that support stage-specific biological functions during Colorado potato beetle development. The embryonic network emphasizes pathways involved in growth regulation and metabolism, while the adult network shows enrichment in signaling pathways related to mature physiological functions.

### 3.3. Identification and Functional Characterization of Tissue-Specific Genes

To identify tissue-specific genes, we calculated the *Tau* index for 15,009 genes across all examined tissues. The distribution of *Tau* index values showed a right-skewed pattern with a prominent peak around 0.8, extending to 1.0 (Figure 3A). This distribution suggests that a substantial proportion of genes exhibit moderate to high tissue specificity, with fewer genes showing ubiquitous expression.

Using a stringent threshold (*Tau* ≥ 0.9), we identified 3616 tissue-specific genes (Figure 3B). Notably, the testis exhibited the highest number of tissue-specific genes (*n* = 1182), greatly exceeding all other tissues. Several other tissues, including ovaries (*n* = 305), hemolymph (*n* = 291), aedeagus (*n* = 282), larval midgut (*n* = 268), genital ducts (*n* = 264), and larval white fat body (*n* = 161), also exhibited tissue-specific genes. However, these numbers were much lower than those in the testis.

Given the remarkable abundance of testis-specific genes, we performed detailed functional annotation of these genes. GO enrichment analysis of testis-specific genes showed significant enrichment of biological processes primarily related to reproductive functions and cellular division (Figure 3C). The most significantly enriched terms included microtubule-based processes, fertilization, sperm motility, detection of stimulus, cell division, and meiotic cell cycle processes (Figure 3C). At the cellular component level, these genes were significantly associated with microtubule-associated complexes, ciliary structures, spindle midzones, and replication forks (Figure 3C). Molecular function analysis showed enrichment in microtubule motor activity, ATP hydrolysis, DNA-dependent ATPase activity, and transmembrane transporter activity (Figure 3C). The protein–protein interaction network identified several highly interconnected hub genes, including NAPA, MAPK15, PRCP, and LAP3, which showed extensive connectivity with many testis-specific genes (Figure 3D). This pronounced network centrality suggests these hub genes may function as master regulators orchestrating testis-specific biological processes.

### 3.4. Analysis of mRNA Sequence Polymorphisms Across Developmental Stages and Tissues

To examine transcriptome sequence variations throughout development and across tissues, we analyzed nucleotide polymorphisms, including both single nucleotide polymorphisms (SNPs) and insertions/deletions (InDels).

The frequency distribution of sequence variants per transcriptional unit revealed a pronounced negative exponential pattern (Figure 4A). A total of 43% of all mutated genes exhibited fewer than 10 sequence variants, while 25.9% contained between 10 and 20 variants (Figure 4A). The proportion of genes decreased exponentially with increasing variant frequency, with only 1.6% containing more than 100 variants (Figure 4A). This skewed distribution suggests strong purifying selection preserving sequence fidelity in most transcripts, while a small subset exhibits elevated polymorphism rates. Variant-containing genes showed a negative correlation between GC content and transcript length (log10-transformed), with longer genes exhibiting lower GC composition (Figure 4B). Density distributions reveal that variant-containing genes are generally longer and show greater GC content variation than invariant genes, suggesting sequence context-dependent mutational patterns.

The highest SNP densities were detected in fertilized egg transcriptomes, followed by progressively decreasing densities through early larval stages and later developmental periods (Figure 4C). Among differentiated tissues, the gut and reproductive organs showed higher SNP frequencies compared to other tissues. Hemolymph exhibited the lowest SNP density among all examined tissues (Figure 4C), suggesting stronger selective constraints on transcript sequence integrity in terminally differentiated tissues. Similarly, the distribution of InDels across developmental stages demonstrated a parallel pattern to SNP distribution, with maximal frequencies observed in fertilized egg transcriptomes, which gradually decreased through larval development (Figure 4D). In adult tissues, the female midgut, male fat body, and genital ducts exhibited stable, moderate InDel frequencies, while tissues like the hindgut, Malpighian tubules, and aedeagus displayed intermediate densities (Figure 4D). Although the hemolymph showed the lowest SNP density among all tissues, its InDel frequency remained comparable to that of other adult tissues, including the male and female gonads (Figure 4C,D).

These results highlight the dynamic nature of transcriptome sequence variation during development, with early stages exhibiting higher frequencies of both SNPs and InDels.

### 3.5. Identification of P450 Genes in Colorado Potato Beetle

Cytochrome P450 monooxygenases represent a critical molecular foundation for insect adaptation to complex environments. To address the lack of systematic characterization of P450 genes in the Colorado potato beetle genome, we performed a comprehensive genome-wide screening and phylogenetic analysis. Our investigation identified 78 P450 genes that clustered into four major clans: Clan 2 (*n* = 9), Clan 3 (*n* = 39), Clan 4 (*n* = 20), and the mitochondrial clan (*n* = 10) (Figure 5A). These genes were unevenly distributed across all 18 chromosomes, with chromosomes 1, 2, and 3 showing higher gene density (Figure 5B). We observed distinct gene clusters in specific chromosomal regions, suggesting potential tandem duplication events during evolutionary history.

### 3.6. Expression Profiles of P450 Genes in Colorado Potato Beetle

To investigate the functional specialization of the identified P450 genes, we analyzed their expression patterns across various developmental stages and tissues in the Colorado potato beetle.

The P450 genes were grouped into six distinct clusters based on their expression profiles (Figure 6). Cluster 1, with the largest number of P450 genes (*n* = 21), showed predominant expression during early larval and embryonic stages. Cluster 5 contained 16 genes with high tissue-specific expression in the fat body. Cluster 2 (*n* = 12) was enriched in the hindgut tissue, while Cluster 6 (*n* = 12) exhibited elevated expression during adult stages. The genes in Cluster 3 (*n* = 10) were characterized by high expression levels in both the Malpighian tubules and midgut tissues. Finally, Cluster 4, with seven genes, showed tissue-specific enrichment in the midgut and lower expression in the Malpighian tubules. These expression patterns reveal the functional specialization of P450 genes across different developmental stages and tissues in the Colorado potato beetle.

### 3.7. Evolutionary Selection Pressure of P450 Genes in Colorado Potato Beetle

To identify potential adaptive evolution in P450 genes, we analyzed selection pressure by calculating nonsynonymous to synonymous substitution ratios (dN/dS). Five of the examined P450 genes (*CYP15C1*, *CYP6BQ11*, *CYP6P4*, *CYP4BN6*, and *CYP9Z26*) exhibited signatures of positive selection (Figure 7). Four genes showed highly significant positive selection (*p* = 0): *CYP15C1* (maximum dN/dS = 24,440), *CYP6BQ11* (100,000), *CYP6P4* (5725), and *CYP4BN6* (231.9). *CYP9Z26* also displayed positive selection (*p* = 0.03), with a maximum dN/dS of 14.72 (Figure 7).

The proportion of sites under selection pressure varied among these genes. Most sites in all five genes showed dN/dS values below 1, suggesting predominant purifying selection. However, sites with extremely high dN/dS ratios indicate positions under strong positive selection, possibly linked to adaptive functions in the Colorado potato beetle.

### 3.8. Tissue and Developmental Stage-Specific Mutation Patterns in P450 mRNA Transcripts

To characterize mRNA mutation patterns in cytochrome P450 genes, we analyzed SNP and InDel frequencies across different tissues and developmental stages in the Colorado potato beetle.

Several P450 transcripts showed enrichment of SNPs specific to certain tissues or life stages (Figure 8A). High SNP frequencies were observed in transcripts from the fat body (both male and female) and midgut tissues, with distinctive patterns across different larval instars (L1, L2, and L3). Fertilized egg samples also displayed distinctive SNP profiles for several P450 transcripts.

The InDel distribution in mRNA showed patterns that were partially distinct from the SNP distribution (Figure 8B). Some P450 transcripts displayed particularly high InDel frequencies in reproductive tissues (ovaries, testis) and digestive organs (midgut). Developmental stage-specific patterns were also observed, with some P450 transcripts showing elevated InDel frequencies during larval instars. The highest InDel frequencies were observed in a subset of P450 transcripts from specific tissues and developmental stages.

These mRNA mutation patterns across tissues and developmental stages suggest variations in P450 transcript processing and stability, potentially reflecting tissue- and stage-specific transcriptional regulation and RNA editing of P450 genes in the Colorado potato beetle.

## 4. Discussion

### 4.1. Differential Expression Analysis Reveals Gene Expression Dynamics During Key Developmental Transitions in Colorado Potato Beetle

Our differential expression analysis clearly revealed the dynamic patterns of gene expression throughout Colorado potato beetle development. The significant variations in the number of differentially expressed genes (DEGs) between adjacent developmental stages, especially during the egg-to-L1 and L3-to-adult transitions, highlight these as critical developmental transition points.

The enrichment of polysaccharide digestion genes in L1 versus egg comparisons likely indicates the initiation of feeding behavior, critical for Colorado potato beetle adaptation to its specialized Solanaceous plant diet [1]. The L1-to-L2 and L2-to-L3 transitions showed fewer gene expression changes, while the L3-to-adult transition displayed the most extensive reprogramming (359 upregulated and 511 downregulated genes). This substantial change during the L3-to-adult transition is consistent with extensive tissue remodeling during metamorphosis, where differential expression of ecdysteroid metabolism genes highlights these hormones’ central role in regulating the metamorphic process [47]. Throughout development, genes associated with chitin-based cuticle development were differentially expressed, reflecting periodic molting and cuticle remodeling [48].

### 4.2. Stage-Specific Gene Expression Patterns Reveal Distinct Developmental Specialization

Embryonic stage-specific genes were most abundant (*n* = 1969), followed by adult-specific genes (*n* = 1647), with fewer stage-specific genes in the larval stages (L1, L2, and L3). This pattern mirrors findings in *Drosophila*, indicating higher transcriptome specialization in the embryonic and adult stages compared to the larval stages [49].

The enrichment of circadian entrainment, fatty acid elongation, amino sugar and nucleotide sugar metabolism, mTOR signaling, and the Hippo signaling pathway during the embryonic stage is consistent with early developmental regulation. These critical pathways show distinctive enrichment in the embryonic network, indicating their decisive role during this key period. These pathways are critical for embryonic development in other organisms, such as the Hippo signaling pathway’s role in controlling organ size and cell proliferation [50], and the mTOR pathway’s role in coordinating growth and metabolism [51].

In contrast, the adult gene network was enriched in cytokine–cytokine receptor interaction, MAPK signaling, thyroid hormone signaling, RAP1 signaling, and various metabolic pathways, reflecting the unique molecular mechanisms required for adult physiological functions. These differences highlight transcriptional reprogramming in the Colorado potato beetle across life cycle stages [47,52]. Transcriptional specialization in embryonic and adult stages may reflect selective pressures during different life history stages, contributing to our understanding of how the Colorado potato beetle adapts to its ecological niche.

### 4.3. Tissue-Specific Gene Expression Reveals Transcriptomic Specialization in Reproductive Tissues

Tissue specificity analysis revealed significant transcriptional differences across tissues in the Colorado potato beetle. We identified 3616 tissue-specific genes, with the testis exhibiting the highest number (*n* = 1182), far exceeding other tissues. Functional annotation of testis-specific genes revealed associations with microtubule-based processes, fertilization, sperm motility, and cell division, aligning with the primary functions of spermatogenesis and sperm maturation [53].

Protein–protein interaction network analysis identified several highly interconnected hub genes, including NAPA, MAPK15, PRCP, and LAP3, which may function as master regulators orchestrating testis-specific biological processes. For example, MAPK15 has been shown in other species to be involved in regulating cell division and differentiation [54], which is essential for spermatogenesis. These hub genes may serve as potential targets for future studies aimed at developing reproductive control strategies for the Colorado potato beetle.

Other tissues, such as the ovaries, hemolymph, and larval midgut, also exhibited significant numbers of tissue-specific genes, indicating unique functional specialization in each tissue. Identifying these tissue-specific genes provides valuable insights into tissue function and development in the Colorado potato beetle and may contribute to more precise pest management strategies.

### 4.4. Developmental Stage Influences Transcriptome Sequence Polymorphism

Our analysis of mRNA sequence polymorphisms revealed intriguing patterns across developmental stages and tissues. The negative exponential distribution of sequence variants per gene, with most genes harboring fewer than 20 variants, suggests strong purifying selection maintaining sequence fidelity for most transcripts [55].

Elevated SNP and InDel frequencies in egg transcriptomes and early larval stages, compared to differentiated tissues and later developmental stages, suggest reduced selective constraints during early development [56]. This pattern may reflect greater tolerance for genetic variation during early developmental stages, where compensatory mechanisms may buffer against deleterious effects of mutations [57]. Alternatively, this could indicate more active genetic processes such as alternative splicing or RNA editing during early development. The reduced polymorphism in hemolymph and other differentiated tissues suggests increased selective pressure to maintain precise transcript sequences in tissues with specialized functions.

### 4.5. Evolutionary Selection Pressure of P450 Genes

The selection pressure analysis of P450 genes reveals a complex evolutionary landscape characterized by predominantly purifying selection punctuated by sites under intense positive selection. This evolutionary pattern—conservation of essential enzymatic function while allowing adaptation at key positions—suggests a fine balance between functional integrity and adaptive capability [58]. Genes under strong positive selection likely play crucial roles in recent adaptive responses to environmental challenges. These rapidly evolving sites may represent functional hotspots involved in substrate recognition or catalytic activity that enable the detoxification of novel compounds. The five P450 genes identified under positive selection represent different clans with distinct evolutionary origins and functional specializations, reflecting the diverse selective pressures faced by the Colorado potato beetle. Based on phylogenetic classification, these genes belong to three major P450 clans: Clan 2 (*CYP15C1*), Clan 3 (*CYP6BQ11*, *CYP6P4*, and *CYP9Z26*), and Clan 4 (*CYP4BN6*), each with characteristic functional roles in insect physiology and adaptation [59,60].

*CYP15C1*, belonging to Clan 2, represents one of the most conserved P450 lineages involved in essential physiological processes, particularly juvenile hormone biosynthesis [61,62]. The extreme positive selection observed in *CYP15C1* (dN/dS = 24,440) is particularly striking given the highly conserved nature of Clan 2 enzymes, suggesting intense selective pressure on developmental timing and reproductive processes [63]. This evolutionary signature may indicate adaptation in metamorphic timing, reproductive synchronization, or stress-responsive developmental plasticity, which could be crucial for the Colorado potato beetle’s ability to rapidly adapt to changing agricultural environments and host plant availability [64,65].

The three Clan 3 genes (*CYP6BQ11*, *CYP6P4*, and *CYP9Z26*) under positive selection represent the primary xenobiotic detoxification machinery in insects [66,67]. CYP6 subfamily members are particularly notable for their role in metabolizing a broad spectrum of xenobiotics, including organophosphates, carbamates, pyrethroids, and neonicotinoids [68,69]. The positive selection likely represents adaptive mutations that enhance substrate binding affinity, catalytic efficiency, or enzyme stability, enabling more effective detoxification of synthetic compounds [70,71].

*CYP9Z26* belongs to the CYP9 subfamily, which is traditionally associated with the metabolism of plant secondary metabolites [72]. The evolution of this gene may reflect ongoing adaptation to the diverse alkaloids and glycoalkaloids present in solanaceous host plants, including α-solanine and α-chaconine in potato plants [73]. This adaptation could enable the beetle to exploit a broader range of potato varieties or to colonize alternative solanaceous hosts with different defensive compound profiles.

*CYP4BN6*, belonging to Clan 4, typically functions in fatty acid ω-hydroxylation and the biosynthesis of semiochemicals involved in chemical communication [74]. CYP4 clan genes have been linked with insecticide resistance in several insect orders, including Coleoptera [75]. The evolution of *CYP4BN6* may also relate to thermal adaptation, as CYP4 enzymes are involved in the production of cuticular hydrocarbons that affect water balance and thermal tolerance [76]. Given the Colorado potato beetle’s need to survive in diverse climatic conditions across its expanded geographic range, positive selection in this gene could reflect adaptation to varying temperature and humidity regimes encountered in different potato-growing regions.

From a pest management perspective, these findings highlight the rapid evolutionary potential of P450-mediated resistance mechanisms and underscore the need for resistance management strategies that consider the evolutionary dynamics of multiple P450 clans simultaneously [77]. The involvement of genes from different functional categories suggests that integrated pest management approaches should address not only direct insecticidal effects but also consider impacts on developmental timing, chemical communication, and physiological adaptation.

### 4.6. Tissue and Developmental Stage-Specific Mutation Patterns

The tissue and developmental stage-specific mutation patterns in P450 transcripts reveal the dynamic regulation of these detoxification genes. The enrichment of genetic variations in metabolically active tissues, such as the fat body, midgut, and reproductive organs, reflects differentiated selection pressures and regulatory mechanisms. The divergent distribution patterns between SNPs and InDels suggest these genetic variations may serve different functional roles or be subject to distinct selective forces. Stage-specific patterns across larval instars and in fertilized eggs indicate specialized roles for certain P450 genes during development. These transcript variation patterns provide evidence of sophisticated regulatory strategies [78], enabling the beetle to adapt to diverse metabolic challenges and contributing to our understanding of the molecular mechanisms behind this pest’s ecological success.

### 4.7. Limitations of RNA-Based Mutation Analysis

The sequence polymorphisms identified in mRNA transcripts require careful interpretation, as RNA-seq data cannot distinguish between genuine genomic mutations and other sources of sequence variation [79,80]. The variants detected may originate from multiple mechanisms: true genomic polymorphisms that are faithfully transcribed and could contribute to functional diversity [79,80,81]; RNA editing events, particularly A-to-I editing mediated by ADAR enzymes, which remain poorly characterized in Coleoptera [82]; technical artifacts, including sequencing errors and alignment issues, despite stringent quality filtering; or allelic expression differences in heterozygous individuals due to cis-regulatory effects [83]. The biological coherence of the observed patterns—elevated variant frequencies in early developmental stages and specific tissues—suggests that these are not solely technical artifacts. Moreover, the functional context of variants occurring in genes under positive selection supports their potential biological relevance.

Our validation approach aimed to maximize confidence in detected variants while acknowledging RNA-seq limitations. Variants were required to be consistently detected across multiple biological replicates, and the tissue-specific patterns observed align with expected adaptive evolution patterns in response to environmental pressures. Regardless of their precise molecular origins, these transcript variations carry significant biological implications: amino acid-changing variants could alter protein properties, while even synonymous variants may influence mRNA stability, translation efficiency, or regulatory functions [84,85]. The tissue-specific and developmental stage-specific patterns suggest complex, context-dependent gene regulation mechanisms contributing to organismal adaptability. However, a definitive distinction between genomic mutations and RNA-specific modifications requires paired DNA–RNA analysis from the same individuals, which was beyond our current scope.

Despite these limitations, our findings provide valuable insights into gene regulation dynamics across developmental stages and tissues. The patterns of sequence variation we identified highlight the multi-layered complexity of transcriptional regulation, whether reflecting genomic diversity, post-transcriptional modifications, or regulatory differences. These results establish a foundation for future mechanistic studies incorporating whole-genome sequencing alongside transcriptomic profiling to resolve the molecular basis of observed polymorphisms. Our work underscores the importance of cautious interpretation when studying adaptive gene families through transcriptomic approaches while demonstrating that such analyses can still provide meaningful insights into the molecular mechanisms underlying organismal adaptability and inform the development of targeted research strategies.

## 5. Conclusions

This study provides a comprehensive transcriptome analysis of the Colorado potato beetle across developmental stages and tissues, focusing on the P450 gene family. We identified stage-specific and tissue-specific gene expression patterns, with embryonic, adult, and testis tissues showing the highest levels of transcriptional specialization. Genome-wide screening revealed 78 P450 genes distributed across four major clans. The uneven chromosomal distribution and clustering patterns suggest that tandem duplication is a key evolutionary mechanism driving P450 family expansion. Expression analysis identified six distinct clusters with tissue- and developmental stage-specific patterns, indicating functional specialization throughout the beetle’s life cycle. Evolutionary analysis highlighted five P450 genes under strong positive selection, suggesting their involvement in adaptive responses to environmental challenges. Additionally, analysis of mRNA sequence polymorphisms revealed developmental stage- and tissue-dependent mutation patterns, with elevated mutation frequencies in early developmental stages and specific tissues like the fat body and midgut. These patterns reflect differentiated selection pressures and regulatory mechanisms, which likely contribute to the beetle’s remarkable ecological success. Our findings advance the understanding of the molecular mechanisms underlying the beetle’s adaptability and provide valuable targets for developing novel pest management strategies to combat this devastating agricultural pest.

## Figures and Tables

**Figure 1 insects-16-00608-f001:**
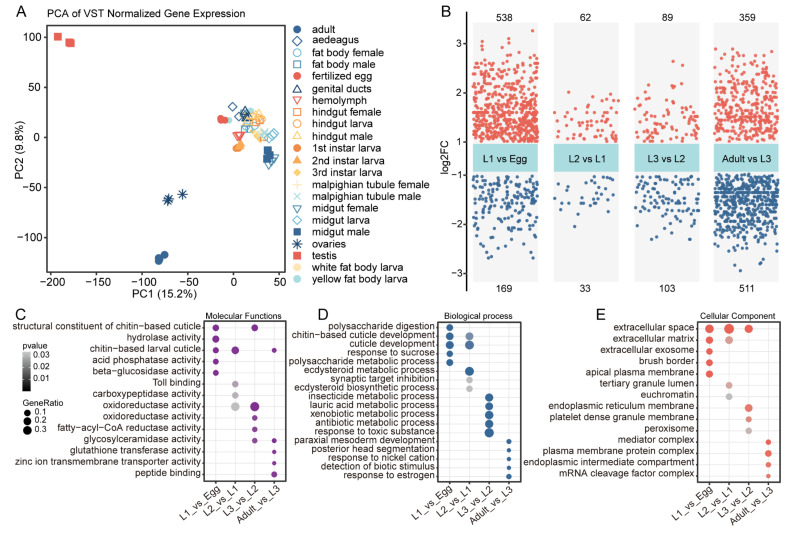
Transcriptome analysis of developmental stages and tissues in the Colorado potato beetle. (**A**) Principal Component Analysis (PCA) of VST-normalized gene expression data from 65 datasets representing different developmental stages and tissues of the Colorado potato beetle. All tissues and developmental stages are represented by at least three biological replicates, except for the larval fat body. (**B**) Differentially expressed genes (DEGs) between consecutive developmental stages. Red dots represent upregulated genes, and blue dots represent downregulated genes (padj ≤ 0.05, |log2FoldChange| ≥ 1). The numbers at the top and bottom of each panel indicate the count of upregulated and downregulated genes, respectively. (**C**–**E**) Gene Ontology (GO) enrichment analysis of DEGs across developmental transitions. (**C**) Molecular Functions, (**D**) Biological Processes, and (**E**) Cellular Components enriched in the DEGs. Dot size represents gene ratio, and color intensity indicates statistical significance (−log10 *p*-value). The analysis reveals stage-specific enrichment of functional categories related to cuticle development, metabolic processes, and cellular structures.

**Figure 2 insects-16-00608-f002:**
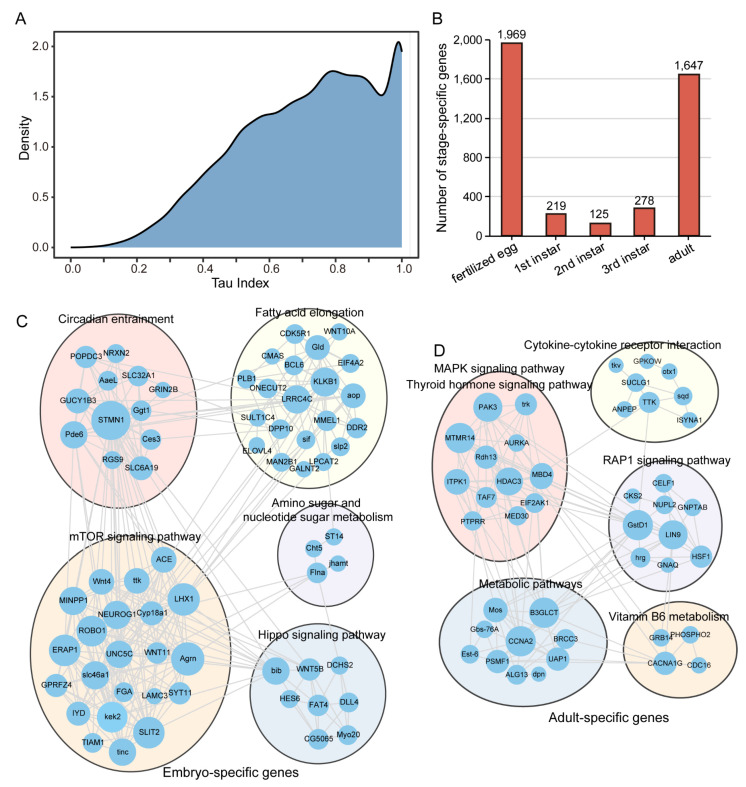
Stage-specific gene expression patterns and associated functional networks. (**A**) Distribution of *Tau* index across developmental stages. (**B**) Number of stage-specific genes (*Tau* ≥ 0.9). (**C**,**D**) Co-expression KEGG networks of embryo and adult-specific genes constructed using Weighted Gene Co-expression Network Analysis (WGCNA). Genes are clustered into functional modules based on their correlated expression patterns. The embryonic network (**C**) shows key pathways including circadian entrainment, fatty acid elongation, amino sugar and nucleotide sugar metabolism, mTOR signaling pathway, and Hippo signaling pathway. The adult network (**D**) reveals distinct functional modules enriched in cytokine–cytokine receptor interaction, MAPK signaling pathway, thyroid hormone signaling pathway, RAP1 signaling pathway, metabolic pathways, and vitamin B6 metabolism. Each circle represents a gene, with connections indicating their co-expression relationships.

**Figure 3 insects-16-00608-f003:**
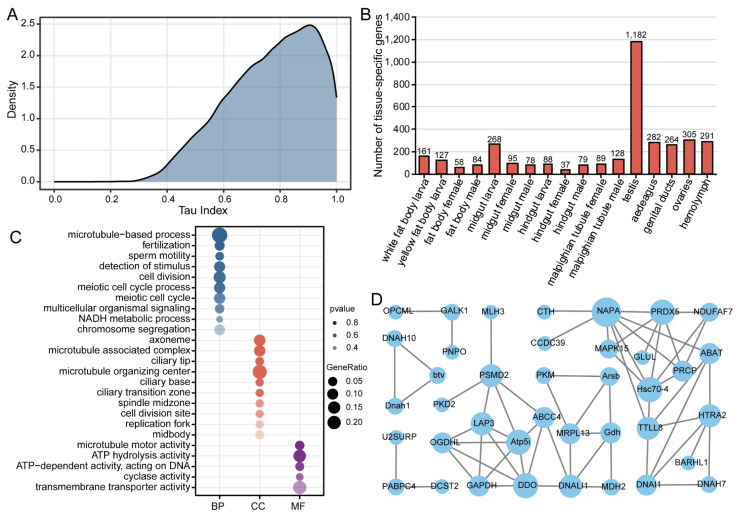
Identification and annotation of tissue-specific genes. (**A**) Distribution of *Tau* index values indicating gene expression specificity. Values closer to 0 represent broadly expressed genes across tissues, while values approaching 1 indicate tissue-specific expression. (**B**) Number of tissue-specific genes (*Tau* ≥ 0.9) across different tissues, with the testis showing the highest number of tissue-specific genes. (**C**) Gene Ontology (GO) enrichment analysis of testis-specific genes. The size of the dots represents the gene ratio, while the color indicates the significance level (*p*-value). BP: biological process; CC: cellular component; MF: molecular function. (**D**) Protein–protein interaction network of testis-specific genes constructed using WGCNA. Node size represents connectivity.

**Figure 4 insects-16-00608-f004:**
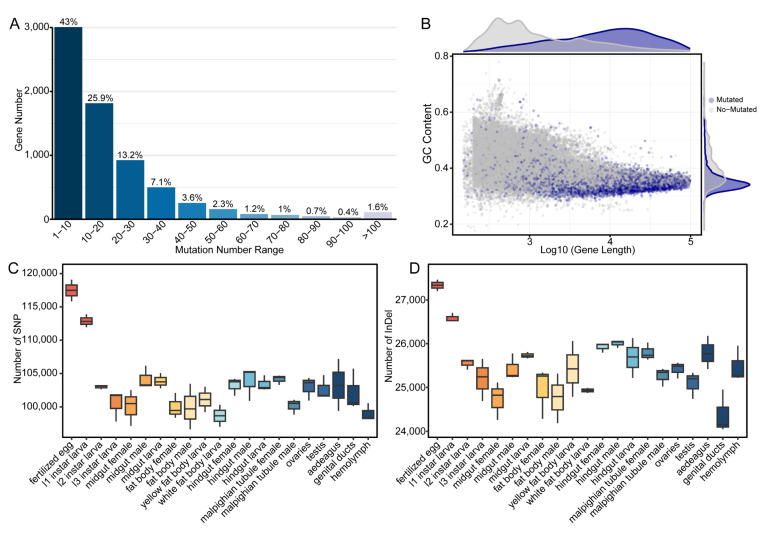
Characterization of nucleotide sequence variations across developmental stages and tissues. (**A**) Frequency distribution of sequence variants per gene, with percentages indicating the proportion of genes within each variant frequency interval. (**B**) Correlation between GC content and gene length (log10-transformed) for variant-containing (blue) and invariant (gray) genes, with corresponding density distributions displayed marginally. (**C**) Quantification of single nucleotide polymorphisms (SNPs) across ontogenic stages and tissue types. (**D**) Distribution of insertions and deletions (InDels) across developmental progression and differentiated tissues. Box plots in panels (**C**,**D**) illustrate median values, interquartile ranges, and statistical outliers for each biological sample.

**Figure 5 insects-16-00608-f005:**
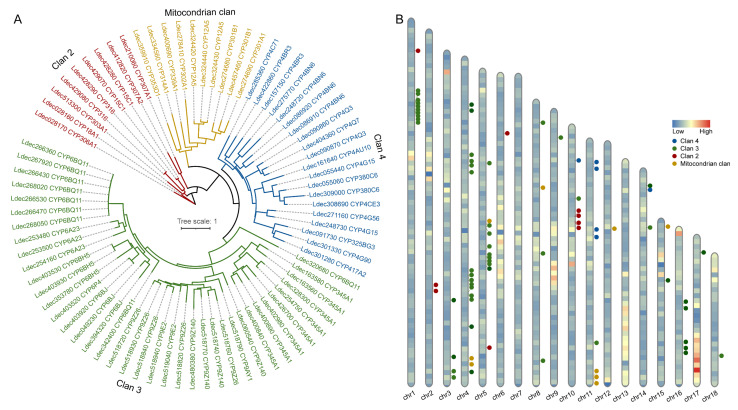
Phylogenetic analysis and chromosomal distribution of Colorado potato beetle P450 genes. (**A**) Phylogenetic tree of potato P450 genes showing classification into four distinct clans: Clan 2 (red), Clan 3 (green), Clan 4 (blue), and the mitochondrial clan (yellow). The tree was constructed based on amino acid sequences using the maximum likelihood method, with a scale of 1. (**B**) Chromosomal mapping of P450 genes across potato chromosomes (chr1–chr18). The distribution pattern indicates gene clusters, with different expression levels represented by a color gradient from low (blue) to high (red). Colored dots represent genes from different clans, as indicated in the legend.

**Figure 6 insects-16-00608-f006:**
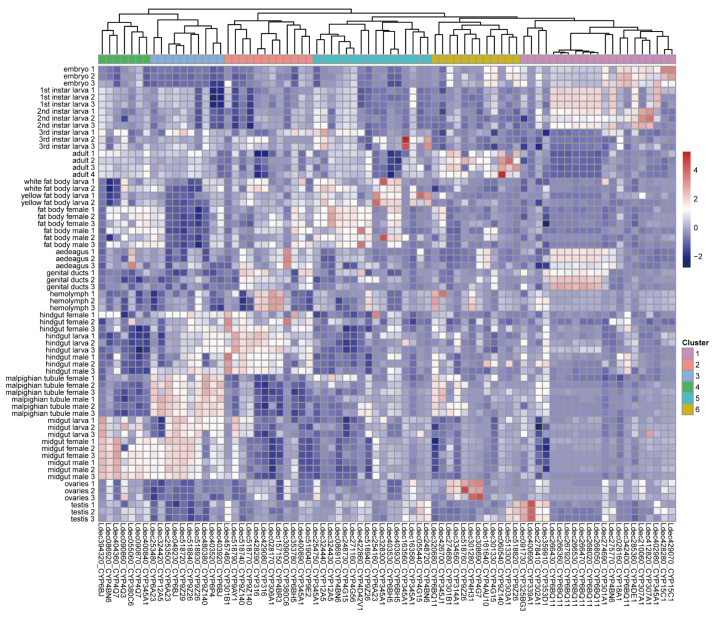
Expression profiles of Colorado potato beetle P450 genes across different developmental stages and tissues. Hierarchical clustering of P450 gene expression patterns from RNA-seq data across multiple developmental stages (embryo, larvae, pupae, adults) and tissues (fat body, midgut, hindgut, Malpighian tubules, ovaries, testes, genital ducts, hemolymph). The heatmap shows normalized expression values represented by a color gradient from −2 to 4.

**Figure 7 insects-16-00608-f007:**
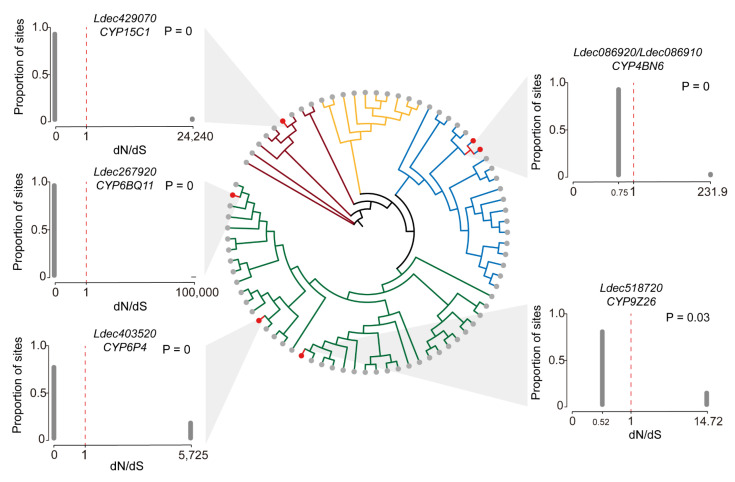
Selection pressure analysis of P450 genes in the Colorado potato beetle. Bar plots illustrate the dN/dS ratios for five representative P450 genes (*CYP15C1*, *CYP6BQ11*, *CYP6P4*, *CYP4BN6*, and *CYP9Z26*). The dashed red lines indicate dN/dS = 1. *p*-values represent the statistical significance of positive selection tests. The proportion of sites under different selection pressures is shown for each gene.

**Figure 8 insects-16-00608-f008:**
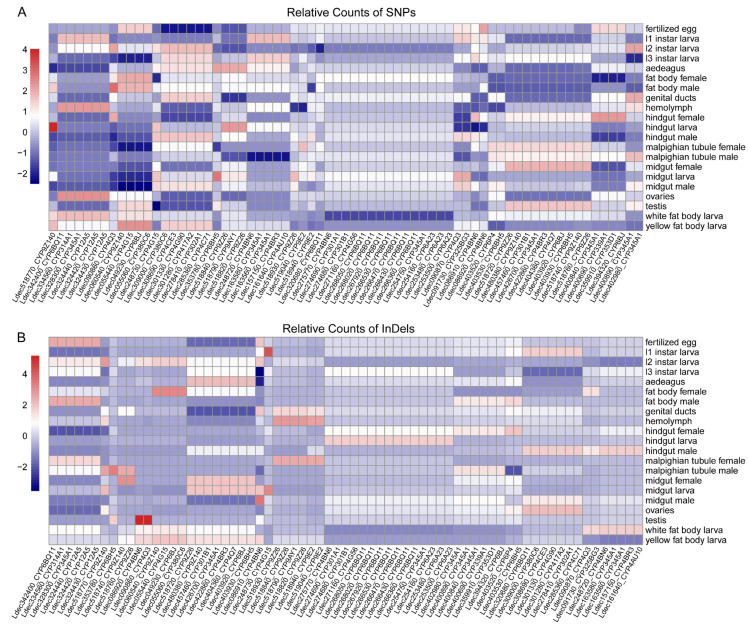
Relative mRNA mutation frequencies in cytochrome P450 genes of the Colorado potato beetle. Heatmaps showing the relative counts of (**A**) SNPs and (**B**) InDels in P450 mRNA transcripts across different P450 genes (x-axis) in various tissues and developmental stages (y-axis). Color scale indicates relative mutation frequency, with red representing high mutation frequency and blue representing low mutation frequency. White cells indicate no data available or mutation counts below the detection threshold.

## Data Availability

All RNA-seq datasets are available in the NCBI Sequence Read Archive (SRA). The accession numbers for each dataset are provided in Appendix A. The expression matrix and *Tau* index are available in Appendix A.

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
