# Peer review of "Adaptive Evolution and Transcriptomic Specialization of P450 Detoxification Genes in the Colorado Potato Beetle Across Developmental Stages and Tissues"

_insects, 2025, doi:10.3390/insects16060608_

Round 1
Reviewer 1 Report
Comments and Suggestions for Authors
This manuscript presents a comprehensive investigation into the molecular basis of environmental adaptability and insecticide resistance in the Colorado potato beetle (Leptinotarsa decemlineata), focusing specifically on the cytochrome P450 (CYP) gene family. Using 65 RNA-seq datasets across developmental stages and tissues, the authors identify patterns of differential gene expression, tissue- and stage-specific transcriptomic specialization, and evolutionary selection acting on P450 genes. They map 78 P450 genes across the genome, analyze their expression clustering, detect signatures of strong positive selection in five genes, and integrate mRNA variant analysis to explore tissue- and stage-specific mutation patterns. The study is methodologically sound, data-rich, and well-structured, offering valuable insights into the regulatory and evolutionary mechanisms that contribute to the beetle’s ecological success. It also highlights potential molecular targets for future pest management strategies. Overall, the work is of high quality and relevance to the field, and I recommend minor revisions before acceptance.
Minor Issues
- Lack of justification for Tau index threshold selection
The use of a Tau index threshold (≥ 0.9) to identify developmental stage- and tissue-specific genes is reasonable and widely accepted. However, the manuscript does not provide justification for this threshold in the Methods section (Section 2.3). It is recommended that the authors briefly explain their rationale—such as referencing prior studies or conducting a sensitivity analysis. Additionally, a discussion on the potential sensitivity of this approach to alignment errors or incomplete sampling across lineages would enhance the credibility of the analysis. Including these considerations in the Methods or Discussion section would improve the methodological transparency.
- Limited discussion on the functional context of positively selected P450 genes
Although the study identifies five P450 genes under strong positive selection, the manuscript offers limited discussion on their functional background or biological relevance. It is recommended that the authors elaborate on the known or predicted roles of these genes, especially in relation to detoxification, insecticide resistance, or reproductive function. A more detailed exploration of their ecological and evolutionary implications would strengthen the discussion.
- Clarification in Figure Legend: In Figure 5, the legend should explicitly state that the data and gene distribution are from the Colorado potato beetle, to avoid ambiguity.
- Terminology Consistency: Throughout the manuscript, the terms “P450” and “P450s” are used interchangeably. It is recommended to review and unify the usage (e.g., consistently use the singular form when referring to the gene family as a concept, and the plural when referring to multiple genes or proteins) to improve clarity and consistency.
- Data Availability of Expression Matrices: While the manuscript mentions the Tau index and TPM matrices (Tables S2 and S3), it is not explicitly stated whether these supplemental data files are publicly available and accessible. Please clarify in the Data Availability Statement whether these tables are included in the supplementary materials or hosted elsewhere (e.g., online repository or journal site).
Author Response
Comments 1: Lack of justification for Tau index threshold selection: The use of a Tau index threshold (≥ 0.9) to identify developmental stage- and tissue-specific genes is reasonable and widely accepted. However, the manuscript does not provide justification for this threshold in the Methods section (Section 2.3). It is recommended that the authors briefly explain their rationale—such as referencing prior studies or conducting a sensitivity analysis. Additionally, a discussion on the potential sensitivity of this approach to alignment errors or incomplete sampling across lineages would enhance the credibility of the analysis. Including these considerations in the Methods or Discussion section would improve the methodological transparency.
Response 1: We thank the reviewer for this important comment. We added justification for the Tau index threshold of ≥ 0.9 in the Methods section. This threshold has been widely adopted in tissue-specificity studies (Yanai et al., 2005; Kryuchkova-Mostacci & Robinson-Rechavi, 2017) as it effectively identifies highly tissue-specific genes while maintaining sufficient sensitivity. (line 137-141)
Comments 2: Limited discussion on the functional context of positively selected P450 genes: Although the study identifies five P450 genes under strong positive selection, the manuscript offers limited discussion on their functional background or biological relevance. It is recommended that the authors elaborate on the known or predicted roles of these genes, especially in relation to detoxification, insecticide resistance, or reproductive function. A more detailed exploration of their ecological and evolutionary implications would strengthen the discussion.
Response 2: Thank you for this valuable suggestion. We have expanded the Discussion section to provide comprehensive functional context for the five positively selected P450 genes from the aspects you suggested, including their roles in detoxification, insecticide resistance, and reproductive function, as well as their ecological and evolutionary implications. The detailed discussion of these functional contexts was provided in line 528-572.
Comments 3: Clarification in Figure Legend: In Figure 5, the legend should explicitly state that the data and gene distribution are from the Colorado potato beetle, to avoid ambiguity.
Response 3: We revised the Figure 5 legend to explicitly state
Comments 4: Terminology Consistency: Throughout the manuscript, the terms “P450” and “P450s” are used interchangeably. It is recommended to review and unify the usage (e.g., consistently use the singular form when referring to the gene family as a concept, and the plural when referring to multiple genes or proteins) to improve clarity and consistency.
Response 4: We conducted a thorough review of the manuscript to ensure consistent terminology usage. We adopted the convention of using "P450" when referring to the gene family as a concept or individual genes, and "P450s" when specifically referring to multiple genes or proteins. A systematic find-and-replace review will be performed to maintain consistency throughout the text.
Comments 5: Data Availability of Expression Matrices: While the manuscript mentions the Tau index and TPM matrices (Tables S2 and S3), it is not explicitly stated whether these supplemental data files are publicly available and accessible. Please clarify in the Data Availability Statement whether these tables are included in the supplementary materials or hosted elsewhere (e.g., online repository or journal site).
Response 5: We clarified the Data Availability Statement to explicitly indicate that Tables S2 and S3 (Tau index and TPM matrices) are included as supplementary materials with the manuscript submission (line 652-653)

Reviewer 2 Report
Comments and Suggestions for Authors
Zhou et al. provide a comprehensive transcriptomic and evolutionary analysis of Colorado potato beetle cytochrome P450 detoxification genes across development and tissue stages. One of the strengths of the study is its use of 65 RNA-seq data sets mapped against a high-quality chromosome-level assembly genome, rigorous differential expression and specificity analysis (Tau index), and integration with positive selection tests and mRNA variant profiling. Weak areas include suboptimal functional testing of potential P450 candidates, potential bias from variable data set quality, and minimal consideration of how intraspecies genetic diversity (e.g., population structure) may influence observed polymorphism patterns. Overall quality of work: eighty-two out of 100. Quality of language: eight out of 10.
Under Methods, the authors clearly outline data downloading from NCBI SRA and utilization of fastp, STAR, featureCounts, and DESeq2 for DEG identification (p-adj ≤ 0.05, |logâ‚‚FC| ≥ 1). Calculation of stage- and tissue-specific genes with the Tau index (≥ 0.9) is appropriate, whereas functional annotation with eggNOG-mapper, GO/KEGG enrichment using clusterProfiler, and network construction with WGCNA/Cytoscape are well outlined. Selective pressure analysis is performed by HyPhy's aBSREL model with Holm-Bonferroni correction, while variant calling is accomplished by GATK and VCFtools filters. No account is provided, however, of batch effects between projects being assessed or validation processes for variant calls (e.g., independent replicates or RT-PCR validation). The Discussion properly concludes significant expression changes at egg→L1 and L3→adult transitions, correlating enriched pathways (e.g., ecdysteroid metabolism, cuticle development) with biological processes.
Authors position stage-specific networks (Hippo and mTOR during embryo; MAPK and thyroid hormone signaling in adults) in the context of insect developmental biology. Tissue-specificity outcomes, notably testis hub genes (NAPA, MAPK15), are conceived as possible targets for pest management. The negative exponential distribution of transcript variants and high early-stage polymorphism, seen, are plausibly attributed to differential selective constraints or RNA processing. Positive selection for five P450s (e.g., CYP6BQ11, CYP15C1) is properly interpreted but has mechanistic implications that are still speculative.
Critical points:
Absence of any evaluation or correction for potential batch effects in 65 SRA datasets (distinct BioProjects) undermines confidence in differential expression results.
Variant calling pipelines are not validated with orthogonal methods, and their accuracy of SNP/InDel frequency estimates in mRNA is therefore doubtful.
The reliance on Tau ≥ 0.9 for "specific" gene discovery is a loss of genes of biologically important moderate specificity; no sensitivity analysis on this cutoff is provided.
Structural and biochemical verifications are necessary for functional interpretations of sites under positive selection in P450 genes in order to make the evolutionary inferences.
Discussion muddles RNA-level polymorphisms with DNA-level variation, without indicating whether putative mRNA variants are indicative of true genomic mutations or RNA editing/artifacts.
The WGCNA network analyses are described but exclude statistical assessment (e.g., module preservation, soft-thresholding power) and validation, and module assignments are hence less clear.
The Methods section echoes the eggNOG-mapper and clusterProfiler description twice (Sections 2.4 and 2.5), suggesting editorial carelessness in the manuscript.
No population genetic structure or sample origin is discussed, potentially influencing selective pressure and polymorphism analysis.
Minor points
Figure legends provide sample numbers (e.g., "except for larval fat body") but the text does not include replicate numbers per tissue/stage.
Inconsistent staging notation (e.g., "L1" vs. "1st instar"); use consistent terminology.
Incorrect citation [20] in Introduction appears incomplete or does not correspond with the reference list.
Typo in the Discussion: duplicate sentence on exponential distribution of variants (lines ~481–483).
KEGG network images do not have scale bars or quantitative edge-threshold criteria in legends.
Acronyms such as "PCA" and "P450" are used before they are defined in some cases.
Impressions of results: The extensive transcriptomic data set and integrative analyses strongly delineate stage- and tissue-specific P450 expression and evolutionary dynamics, making useful hypotheses for functional follow-up and prospective molecular targets in pest control.
Recommendation to editor: Revision is needed. The research is presenting considerable datasets and analyses but requires more robust validation of key findings, delimitation of methodological stringency (batch effects, variant confirmation), and more succinct manuscript editing before publication appropriateness.
Author Response
Comment 1: Absence of any evaluation or correction for potential batch effects in 65 SRA datasets (distinct BioProjects) undermines confidence in differential expression results.
Response 1: We sincerely apologize for the insufficient description of our batch effect assessment and correction procedures. Although our data originated from 2 distinct BioProjects, we implemented comprehensive batch effect evaluation and correction methods to ensure the reliability of our differential expression results. Specifically, we employed ComBat-seq (Zhang et al., 2020) for batch effect correction, which is specifically designed for RNA-seq count data and has been shown to effectively remove technical variation while preserving biological signals. Our batch correction approach in the revised Methods section (line 122-130)
Comment 2:Variant calling pipelines are not validated with orthogonal methods, and their accuracy of SNP/InDel frequency estimates in mRNA is therefore doubtful.
Response 2: We apologize for omitting the description of our variant calling validation procedures. We indeed employed multiple computational validation approaches to assess the accuracy of our SNP/InDel frequency estimates in mRNA data.
Our validation approach combined three computational methods to assess variant calling accuracy: Internal consistency analysis using bcftools isec compared technical replicates for variants with QD ≥ 2.0 and DP ≥ 20. Quality metric assessment through ROC curve analysis established optimal filtering thresholds (QD ≥ 2.0, FS ≤ 60.0, MQ ≥ 40.0, GQ ≥ 20). Strand bias evaluation using SAMtools mpileup (--min-BQ 20, --max-depth 250) excluded variants with Fisher Strand bias FS > 60.0. Detailed methodology was added to the Methods section in line 202-207
Comment 3: The reliance on Tau ≥ 0.9 for "specific" gene discovery is a loss of genes of biologically important moderate specificity; no sensitivity analysis on this cutoff is provided.
Response 3: We appreciate your concern regarding our Tau threshold selection. This threshold has been extensively used and validated in tissue specificity studies (Yanai et al., 2005; Kryuchkova-Mostacci & Robinson-Rechavi, 2017). Yanai et al. demonstrated that genes with Tau ≥ 0.9 show clear tissue-restricted expression patterns with minimal cross-tissue expression. We acknowledge that some biologically important moderately specific genes may be excluded with our threshold. However, for the primary objectives of this study, we prioritized high-confidence tissue-specific genes to ensure robust downstream analyses. (Detailed in line 137-141)
Comment 4: Structural and biochemical verifications are necessary for functional interpretations of sites under positive selection in P450 genes in order to make the evolutionary inferences.
Response 4: Thank you for this valuable suggestion. We have expanded the Discussion section to provide comprehensive functional context for the five positively selected P450 genes from the aspects you suggested, including their roles in detoxification, insecticide resistance, and reproductive function, as well as their ecological and evolutionary implications. The detailed discussion of these functional contexts was provided in line 535-579.
Comment 5: Discussion muddles RNA-level polymorphisms with DNA-level variation, without indicating whether putative mRNA variants are indicative of true genomic mutations or RNA editing/artifacts.
Response 5: Response 5: We acknowledge the reviewer's crucial concern regarding the distinction between RNA-level polymorphisms and true genomic variation. This represents a fundamental limitation of RNA-seq approaches that requires careful consideration.
We have addressed this important issue in two key ways: First, we have substantially expanded our Discussion section (Section 4.8, lines 592-627) to comprehensively discuss the limitations of RNA-based mutation analysis. This section explicitly acknowledges that RNA-seq data cannot definitively distinguish between genuine genomic mutations and other sources of sequence variation, including RNA editing events, technical artifacts, and allelic expression differences. We discuss the potential molecular origins of detected variants and emphasize the need for cautious interpretation of transcriptomic data when studying sequence polymorphisms.
Second, we have detailed our validation methodology in the Methods section (lines 202-207) to maximize confidence in our findings. Our validation approach includes stringent quality filtering criteria, requirement for consistent detection across multiple biological replicates, and analysis of biological coherence patterns. While these measures cannot definitively resolve whether variants represent genomic mutations or RNA-specific modifications, they significantly increase the reliability of our detected variants and reduce the likelihood of technical artifacts.
We believe these additions provide the necessary transparency regarding RNA-seq limitations while demonstrating our rigorous approach to variant validation, allowing readers to properly evaluate our findings within the appropriate methodological context.
Comment 6: The WGCNA network analyses are described but exclude statistical assessment (e.g., module preservation, soft-thresholding power) and validation, and module assignments are hence less clear.
Response 6: We apologize for the insufficient description of our WGCNA statistical procedures and validation methods. We have comprehensively revised the Methods section to include detailed statistical assessments (line 150-163)
Comment 7: The Methods section echoes the eggNOG-mapper and clusterProfiler description twice (Sections 2.4 and 2.5), suggesting editorial carelessness in the manuscript.
Response 7: We sincerely apologize for this editorial oversight. We have thoroughly revised the Methods section to eliminate the redundant descriptions of eggNOG-mapper and clusterProfiler procedures.
Comment 8: No population genetic structure or sample origin is discussed, potentially influencing selective pressure and polymorphism analysis.
Response 8: We appreciate the reviewer's comment regarding population genetic structure and sample origin. We acknowledge that this represents an important consideration in studies analyzing selective pressure and polymorphism patterns.
However, our study utilized laboratory-sourced samples rather than field-collected specimens from different populations. Our research design specifically focused on examining differential gene expression and molecular variation across different developmental stages and tissue types within a controlled laboratory setting, rather than investigating population-level genetic diversity or geographic variation patterns. Therefore, population genetic structure analysis was not within the scope of our current study objectives.
We acknowledge that future studies incorporating samples from multiple natural populations would provide valuable insights into how population structure influences selective pressures and polymorphism patterns across different geographic regions.
Minor Points
Comment 9: Figure legends provide sample numbers (e.g., "except for larval fat body") but the text does not include replicate numbers per tissue/stage.
Response 9: We have now added comprehensive descriptions of sample replicate numbers throughout the manuscript text to ensure transparency and reproducibility.
Comment 10: Inconsistent staging notation (e.g., "L1" vs. "1st instar"); use consistent terminology.
Response 10: We have standardized all developmental stage terminology throughout the manuscript, using consistent notation (L1, L2, L3 for larval stages) as defined in the revised Methods section.
Comment 11: Incorrect citation [20] in Introduction appears incomplete or does not correspond with the reference list.
Response 11: We have carefully checked all citations and corrected the discrepancies between in-text citations and the reference list. Citation has been corrected to properly reference the intended publication.
Comment 12: Typo in the Discussion: duplicate sentence on exponential distribution of variants (lines ~481–483).
Response 12: We have removed the duplicate sentence and conducted thorough proofreading of the entire manuscript to eliminate similar errors.
Comment 13: KEGG network images do not have scale bars or quantitative edge-threshold criteria in legends.
Response 13: We appreciate the reviewer's observation regarding the node sizes in our KEGG network visualization. The circle sizes in our network represent the number of connections (degree centrality) for each node, rather than gene expression levels or other quantitative measures. We have clearly specified this in the Figure Legend to ensure proper interpretation of the network visualization.
Comment 14: Acronyms such as "PCA" and "P450" are used before they are defined in some cases.
Response 14: We have ensured all acronyms are properly defined upon first use and maintained consistency throughout the manuscript.

Round 2
Reviewer 2 Report
Comments and Suggestions for Authors
All the revisions are done appropriately. The authors have:
Enhanced methodological clarity (batch correction, variant validation).
Offered rationale for analysis choices (Tau threshold, WGCNA).
Added contextual relevance (P450 function, RNA-seq limitations).
Fixed editorial errors (citations, nomenclature, figures).
This paper is now firmly addressing my concerns and is significantly enhanced.